# Constipation Mitigation by Rhubarb Extract in Middle-Aged Adults Is Linked to Gut Microbiome Modulation: A Double-Blind Randomized Placebo-Controlled Trial

**DOI:** 10.3390/ijms232314685

**Published:** 2022-11-24

**Authors:** Audrey M. Neyrinck, Julie Rodriguez, Bernard Taminiau, Florent Herpin, Patrice D. Cani, Georges Daube, Laure B. Bindels, Nathalie M. Delzenne

**Affiliations:** 1Metabolism and Nutrition Research Group, Louvain Drug Research Institute, UCLouvain, Université Catholique de Louvain, 1200 Brussels, Belgium; 2Fundamental and Applied Research for Animal and Health (FARAH), Faculty of Veterinary Medicine, University of Liège, 4000 Liège, Belgium; 3CEN Nutriment, Groupe CEN, 21000 Dijon, France; 4WELBIO-Walloon Excellence in Life Sciences and Biotechnology, WELBIO Department, WEL Research Institute, 1300 Wavre, Belgium

**Keywords:** rhubarb, middle-aged, constipation, microbiota, butyrate-producing bacteria, *Lachnospiraceae*

## Abstract

Gut microbiota alterations are intimately linked to chronic constipation upon aging. We investigated the role of targeted changes in the gut microbiota composition in the relief of constipation symptoms after rhubarb extract (RE) supplementation in middle-aged volunteers. Subjects (95% women, average 58 years old) were randomized to three groups treated with RE at two different doses determined by its content of rhein (supplementation of 12.5 mg and 25 mg per day) vs. placebo (maltodextrin) for 30 days. We demonstrated that daily oral supplementation of RE for 30 days was safe even at the higher dose. Stool frequency and consistency, and perceived change in transit problem, transit speed and difficulty in evacuating, investigated by validated questionnaires, were improved in both groups of RE-treated volunteers compared to placebo. Higher abundance of *Lachnospiraceae* (mainly *Roseburia* and *Agathobacter*) only occurred after RE treatment when present at low levels at baseline, whereas an opposite shift in short-chain fatty acid (SCFA) levels was observed in both RE-treated groups (increase) and placebo (decrease). Fecal *Lachnospiraceae* and SCFA were positively correlated with stool consistency. This study demonstrates that RE supplementation promotes butyrate-producing bacteria and SCFA, an effect that could contribute to relieving chronic constipation in middle-aged persons.

## 1. Introduction

Chronic constipation is one of the most common gastrointestinal disorders worldwide, with the most recent systematic review and meta-analysis showing a global prevalence of 14% [1,2]. The prevalence increases with age and is almost twice as common in women than men. Constipation is defined as unsatisfactory defecation resulting from infrequent stools, difficult stool passage, or both [2]. It is a term that embraces a spectrum of symptoms related to an individual’s personal dissatisfaction with their fecal evacuation ability [2]. Symptoms include hard stools, excessive straining, infrequent bowel movements, bloating, and abdominal pain, and if such symptoms last for more than 1 month, constipation is considered as chronic [2]. Chronic constipation management remains a clinical challenge, with frequent suboptimal outcomes to a variety of conservative, behavioral, medical, and surgical interventions. Despite the existence of formal diagnostic criteria for constipation disorders (e.g., Rome criteria) [2], the patients and clinicians often diagnose chronic constipation more pragmatically, based on the self-assessment of symptoms relevance.

The etiology and pathophysiology of chronic constipation remain poorly understood and most likely multifactorial. Chronic constipation is a symptom-based disorder that results from dysfunction of the colonic regulation of stool movement, incoordination of the anorectal neuromuscular apparatus, dysfunction of the bidirectional brain–gut axis and other factors like low fiber diet, old age and dehydration [2]. Recently, the gut microbiota has been suggested to play a role in the pathogenesis of chronic constipation. Bacterial metabolites and end-products of fermentation may affect intestinal motor function through the mediators released by the gut immune response, or intestinal neuroendocrine factors [1]. A direct causal relationship of the gut microbiota on gut motility has been demonstrated in mice, where the prolonged whole-gut transit time in germ-free mice was shortened, and colon contractility was increased when colonized with human microbiota [3]. Furthermore, numerous studies have compared the gastrointestinal microbiota composition between chronically constipated subjects and healthy controls. In general, they report lower bifidobacteria and *Bacteroides*, and sometimes a drop of lactobacilli in constipated adults. Other bacteria like *Coprococcus*, *Roseburia* and *Faecalibacterium* can be modulated but those data seem dependent on the methodology for microbiota analysis [1].

Although significantly reducing the quality of life, chronic constipation can hardly be treated [1]. Laxatives are widely available without prescription and, as a consequence, they are commonly used for self-management of constipation by community-dwelling adults [4]. However, satisfaction levels with laxatives are low because the choice is unappropriated. Although laxatives are effective and commonly used in most patients, several side effects often occur, especially with chronic use, which limits their use in the elderly [5]. Many patients seek help from alternative medicinal and/or natural products. Rhubarb is one of the most popular traditional Chinese medicines used to control various diseases including constipation [6]. The dry extract of rhubarb roots, used in our study, is derived from *Rheum palmatum* (also known as *Rheum officinale* or Chinese rhubarb). This anthraquinone-rich crude extract has a long history of herbal usage in traditional Chinese medicine for the treatment of gastrointestinal diseases. Several studies indicated that rhubarb has antioxidant, anti-cancer, anti-inflammatory, anti-allergic and anti-bacterial properties [7]. The short-term use of rhubarb root as a laxative in case of occasional constipation is recognized and well documented in authoritative texts (EMA/HMPC/113701/2019). In that context, the range recommended for adults, elderly and adolescents over 12 years, is 20–30 mg hydroxyanthracene derivatives (calculated as rhein) daily for less than one week.

We have recently shown that rhubarb extract reduced alcohol-induced hepatic inflammation by a mechanism likely involving the modulation of the gut microbiota [8]. We also found that the same rhubarb extract given for 8 weeks improves gut barrier function, reduces obesity, diabetes, inflammation and increases total fecal energy excretion together with specific changes in the gut microbiota composition [7]. Interestingly, rhubarb extract increased the number of feces produced over 24 h after several weeks of rhubarb treatment in mice [8]. There are no studies that have investigated rhubarb supplementation to treat chronic constipation in a controlled clinical trial until now and the involvement of microbiota in the rhubarb effects on constipation has never been explored. In the present study, we investigated the effects of a rhubarb extract used at 2 different doses (double-blind, randomized, placebo-controlled dose-effect study) as a strategy to improve intestinal transit and to relieve the symptoms of constipation in middle-aged people with focus on the role of the gut microbiota. Tablets of rhubarb roots extract were prepared in order to test hydroxyanthracene derivatives (calculated as rhein) at a dose of 12.5 mg and 25 mg per day for one month; the higher dose corresponding to the recommended posology supported by expert opinions and clinical investigations (see *Assessment report on Rheum palmatum L. and Rheum officinale Baillon, radix—*EMA/HMPC/113701/2019).

## 2. Results

### 2.1. Baseline Characteristics of Volunteers

Forty-two subjects were randomized for the study and were assigned into placebo, rhubarb extract at lower dose (Re × 1, 12.5 mg anthraquinone derivatives expressed in rhein) and at higher dose (Re × 2, 25 mg anthraquinone derivatives expressed in rhein) groups with *n* = 14 in each group. Thirteen subjects were analyzed in each group at the end after loss of follow up for three volunteers. The average age was 58 ± 6 y. The majority of the subjects (95%) included in this study were women. At baseline, the groups were similar in terms of sex, age, body mass index, weight, blood pressure, serum transaminases, glycemia, triglyceridemia and cholesterolemia (Table 1). Before intervention, middle-aged subjects had experienced transit disorders for 18 ± 12 y on average and in the previous two weeks they had experienced pain on average on 2.5 ± 3.9 days. This pain was associated with feeling bloated in 44% of subjects and gas in 41%; one of subjects from the Re × 2 group complained of diarrhea or liquid stools. Thirty-nine volunteers had already taken treatments for transit, essentially laxatives of some sort. The average number of daily bowel movements observed in the reference period before taking the products under study was 0.3 ± 0.1 per day, that is, 2.3 ± 0.8 per week whereas stool consistency was 2.6 ± 0.8. For the whole population, the Mental Component Score (MCS) was 52 ± 6 and the Physical Component Score (PCS) was 46 ± 7; both determined according the SF12 questionnaire.

### 2.2. Rhubarb Extract Supplementation Improved Intestinal Transit without Changing Dietary Habits or Safety Parameters

The habitual diet was determined before and after the treatment through a survey (established according to the recommendations of the French National Nutrition and Health Program (PNNS) [9]. No difference in food habits, including water intake vs. other beverages, was detected between groups (Appendix A). Several parameters were analyzed in order to evaluate safety of rhubarb extract supplementation (Table 2). No change was observed in blood pressure, transaminase activities, lipid profile, glycemia, potassium, white blood cells or inflammatory cytokines 30 days after rhubarb extract supplementation whatever the doses considered, except a slight increase in monocyte chemotactic protein-1 (MCP-1) and a slight decrease in hemoglobin at the lower dose.

After the intervention, two of the Rome III diagnostic criteria for constipation -stool frequency and consistency- were improved over time for Re × 2 group vs. placebo; this effect appeared already after 1 week of treatment (Figure 1a,b). Even if the effect was less marked than at the higher dose of rhubarb extract, the lower dose (Re × 1) significantly improved stool consistency after 4 weeks of treatment (Figure 1b). However, the number of days with abdominal discomfort or pain during the last 15 days increased in rhubarb extract groups (+0.2 ± 1.8 days and +2.2 ± 6.3 days for Re × 1 and Re × 2, respectively) whereas it decreased in the placebo group (−2.3 ± 4.3 days). It is important to note that the stool consistency evaluated by the Bristol tool was in the category of “normal transit” since the means ±SD obtained for Bristol score were around or below 5 for both treated groups (4.4 ± 1.2 and 5.3 ± 1.3 for Re × 1 and Re × 2 after 4 weeks of treatment, respectively) and that the number of days with abdominal discomfort or pain was more important in placebo group at the inclusion (3.5 ± 1.3, 1.8 ± 1.2 and 2.1 ± 0.8 days for placebo, Re × 1 and Re × 2, respectively; *p* > 0.05 Kruskal–Wallis test). The proportion of subjects suffering symptoms accompanying abdominal pain or discomfort such as bloating or gas were not significantly affected by the intervention (Appendix A). In contrast to Re × 1 or placebo groups and although the average score obtained to evaluate stool consistency for the Re × 2 group was below 6 considered as “normal transit”, the number of subjects complaining of diarrhea or loose stools increased significantly in the Re × 2 group. Interestingly, 92% of the subjects in the Re × 2 group vs. 62% in the Re × 1 group and 23% in the placebo group, reported a significantly or much improved perception about the transit speed (according to the PGI-I questionnaire) (Figure 1c). Coherently, 77% of the subjects in the Re × 2 group vs. 54% in the Re × 1 group and 23% in the placebo group, reported a significantly or much improved perception about the transit problem (according to the PGI-I questionnaire) (Figure 1d). Satisfaction about evacuating was also improved by the rhubarb extract supplementations (Figure 1e). Intervention led to a general improvement of scores reflecting the mental wellbeing similarly in all groups (Appendix A). We observed that the evolution of PCS scores decreased 30 days after supplementation with the lower dose of rhubarb extract (Re × 1 group) in contrast to the rhubarb extract supplementation with the higher dose or to the placebo groups. Of note, the group Re × 1 exhibited the higher score of PCS and of related physical dimensions at the beginning of the study compared to the other groups. The bodily pain dimension increased in the placebo group. Other scores related to physical wellbeing were improved for both Re × 2 and placebo groups without reaching the significance level Appendix A.

### 2.3. Rhubarb Extract Supplementation Induces Changes in Gut Microbiome

As shown in Figure 2a, there were no statistically significant changes in the alpha diversity plots after 30 days of placebo or rhubarb treatments. However, the β-diversity characterizing the overall gut microbiota composition was modified by the intervention as shown by the principal coordinate analysis of the Bray-curtis (Figure 2b). No changes in the gut microbiota composition were observed at the phylum level whatever the treatment considered (data not shown). At the family level, we observed that the baseline relative abundance for *Eggerthellaceae* and *Desulfitibacteraceae* was different between the groups (*q* = 0.74 and *p* < 0.05, Appendix A). Although *Lachnospiraceae* appeared less abundant in the Re × 1 group than the other groups at baseline, the statistical difference was not reached (Figure 2c and Appendix A). Importantly, the most important change due to rhubarb extract treatment was an increase in *Lachnospiraceae* abundance (+20%) obtained in subjects treated at the lower dose (Re × 1 group); this effect being significant either considering within-group (*q* < 0.05) or between-group statistical analyses (Figure 2 and Appendix A). The Appendix A showed specific changes in the microbiota at the family level and at the genus level (only for taxa significantly affected at *q* value or *p* value). *Lachnospiraceae* was the most abundant bacteria impacted in the Re × 1 group. At the genus level, we observed that several taxa belonging to *Lachnospiraceae* family, i.e., *Lachnospiraceae_ge, Lachnospiraceae_NK4A136_group, Lachnospiraceae_UCG-001*, *Lachnospiraceae_ND3007_group*, *Lachnoclostridium*, *Roseburia* and *Agathobacter*, increased (*p* < 0.05) 30 days after RE rhubarb extract treatment at the lower dose (Re × 1 group); it is important to note that the higher abundance of those bacteria was observed only in the Re × 1 group at day 30 compared to other groups whatever the day considered. Only *Lachnospiraceae_UCG-001* was increased (day 30 vs. day 0) at the higher dose (Re × 2 group). Interestingly, other butyrate-producing bacteria such as *Clostridia* were also increased due to the rhubarb treatment (Re × 1) (*p* < 0.05). In contrast, most of them decreased in placebo due to the intervention (*p* > 0.05; within-group variations analyzed by matched-pairs Wilcoxon signed-rank). Other significant changes at *p* < 0.05 are observed upon rhubarb extract supplementation such as a lower abundance of *Christensenellaceae* and *Oscillospiraceae*.

We analyzed the fecal concentrations of SCFA (Figure 3). We observed that acetate, propionate and butyrate were unaffected by the placebo intervention whereas those 3 SCFA increased due to rhubarb extract supplementations. The increase in acetate, propionate and butyrate at day 30 vs. day 0 reached significance for the Re × 1 group. In addition, between-groups variations analyzed by Kruskal–Wallis test followed by Dunn’s multiple comparisons test demonstrated significant increase in butyrate for Re × 1 group vs. placebo (−8.0 ± 22.0, 27.8 ± 32.6, 38.3 ± 89.6 for placebo, Re × 1 and Re × 2, respectively; *p* < 0.05).

Figure 4 presents correlation analysis between the most important change in fecal bacteria, i.e., change in *Lachnospiraceae* abundance, and parameters that were significantly increased (or improved) due to the rhubarb extract supplementation (correlation analysis was performed considering delta values calculated from day 30 vs. baseline taking into account all subjects). The higher abundance of this bacteria family due to rhubarb extract was positively correlated, almost reaching significance (*p* < 0.1), with fecal acetate, propionate and butyrate and also with the stool consistency. More importantly, fecal SCFA (acetate, propionate, butyrate) were positively and significantly correlated with stool consistency.

## 3. Discussion

Chronic constipation is a prevalent disorder that affects patients’ quality of life and consumes resources in healthcare systems worldwide. Self-management often includes the use of laxative products, most of which may be purchased in pharmacies and elsewhere without prescription [4]. However, failures in the self-management of constipation are frequent and lead to additional costs which add considerably to the financial burden of constipation. Today, there is an increased tendency to use natural herbal remedies. Several medicinal effects have been attributed to the *Rheum* spp. in the traditional and modern medicine such as healing lungs, liver, kidney, womb and bladder diseases, cancer, diabetes, insect bites, relapsing fevers, diarrhea and constipation [10]. The most important chemical structures from rhubarb are anthraquinones, anthrones and phenolic compounds (stilbenes, flavonoids, phenolic glycosides, phenolic acids, cinnamic acid derivatives and tannins) [10]. The effect of different preparations of rhubarb in treating atherosclerosis, acute bleeding of the upper gastrointestinal tract, dysenteric diarrhea, depression and constipation has been demonstrated in some clinical trials as well (for review see [10]). However, the study of rhubarb supplementation to treat chronic constipation has never been investigated in a clinical trial. The primary objective of our randomized, double-blind, controlled clinical trial was to test the influence of a rhubarb extract supplementation at 2 doses in middle-aged healthy subjects with infrequent bowel movements (not severe constipation). The present study reports that stool frequency and consistency improved over time after rhubarb extract supplementation in subjects; this effect appearing to be dose-dependent. More than 90% of the volunteers perceived significantly or much improved of their transit speed in response to the intervention after the rhubarb extract supplementation at the higher dose against 23% for the placebo group. Similarly, nearly 80% and 50% of subjects from Re × 2 and Re × 1 groups, respectively, perceived significantly or much improved of their transit problem and of difficulty in evacuating. However, although their mean average Bristol score was around 5 indicating normal transit, some subjects complained about diarrhea or loose stools with the higher dose. We assessed quality of life using the SF12 survey. This questionnaire allowed us to evaluate the level of 8 health concepts including components of both physical and mental health. Indeed, it was reported a decline in the quality of life related to constipation and bowel-related symptoms has been reported [11,12]. In our study, although mental wellbeing increased with the intervention whatever the treatment, we observed lower score for the physical dimension (PCS) in subjects treated with the lower dose of rhubarb extract (Re × 1 group), probably due to their high physical health of at the beginning of the intervention compared to the other groups (Re × 2 and placebo groups).

Conventional laxatives are associated with electrolyte imbalances [13]. Here, the treatment with rhubarb extract did not affect blood levels of potassium and was safe as no changes in safety parameters were reported after 30 days even at the higher dose. Although we observed higher blood level of MCP-1 in subjects belonging to Re × 1-treated group, values remained in the range of controls [14]. Altogether, this supports that treatment with rhubarb root extract more than one week appears as safe and can be proposed for chronic constipated patients.

Several publications suggest that constipation is associated with gut dysbiosis, and that gut motility can be managed by modulation of gut microbiota [3,15,16,17]. Dysbiosis in chronic constipation (Roma classification II–III criteria) were recently reviewed [18]. Indeed, numerous case–control studies have compared the gut microbiome between chronic constipation and healthy controls. In general, studies in adults report lower bifidobacteria and *Bacteroides* in constipation, with some also reporting lower lactobacilli [2]. In constipated obese children, significant reduction in *Prevotella* and increased abundance of several genera of Firmicutes, including butyrate-producing *Coprococcus*, *Roseburia* and *Faecalibacterium*, were demonstrated based on 16S rRNA gene pyrosequencing [19]. Authors suggested that while it is possible that the observed changes in the microbiome in constipated subjects are a consequence of a low-fiber diet, these changes also predict a different pattern of bacterial fermentation end-products, such as butyrate production, which may contribute to pathogenesis of constipation. More recently, Zhuang et al. have shown that the feces of constipated people are characterized by lower *Faecalibacterium*, *Ruminococcaceae* and *Roseburia* abundance [20]. They also found that gut microbiota characterized with enriched butyrate-producing and depressed *Desulfovibrionaceae* bacteria attenuates constipation symptoms through promoting intestinal hormones secretion and maintaining gut barrier integrity [20]. Few animal studies have explored the effect of rhubarb on the gut microbiota whereas the change of microbiota composition upon rhubarb supplementation has never been investigated in human. We have previously shown that the abundance of *Akkermansia muciniphila*, *Parabacteroides* and *Erysipelatoclostridium* significantly increased upon rhubarb treatment, while *Ruminococcus* and *Peptococcus* significantly decreased in rhubarb-fed mice in a model of high fat/high sucrose-induced obesity [7]. In addition, the most prominent differences upon rhubarb supplementation in a mouse model of binge drinking was related to the *Akkermansia* genus whose relative abundance reached 40% in RE-fed mice after alcohol challenge. Relative abundance of *Alistipes* was increased by alcohol challenge and lowered by RE and *Parabacteroides* was promoted by RE [8]. Beneficial bacteria such as *Ligilactobacillus*, *Limosilalactobacillus*, and *Prevotellaceae* were remarkably increased, and pathogens such as *Escherichia-Shigella* were significantly decreased after rhubarb treatment in a rat model of loperamide-induced constipation [21]. A more recent study mentioned that rhubarb promoted *Alistipes*, *Clostridium*, and *Lactobacillus* proliferation in a mouse model of acute lung injury [22]. In the present study, we have shown that rhubarb extract affected other genera of bacteria and the most important change in the gut microbiota composition observed after rhubarb extract supplementation in constipated subjects was a higher abundance of *Lachnospiraceae* that includes *Agathobacter* and *Roseburia*. Interestingly, although statistical significance was not strictly reached (*p* < 0.1 rather than 0.05), this important family of gut bacteria was positively correlated to the fecal SCFA, including butyrate, and the stool consistency. Importantly, those last parameters correlated also between them, suggesting that butyrate production (among others SCFA) by gut microbiota can manage constipation in middle-aged adult, as previously hypothesized in the study of Zhuang et al. [20] and also by others [23]. In this last study evaluating psyllium husk supplement in 16 constipated patients (average age of 41 years) for 7 days, they demonstrated significant increases in three genera known to produce butyrate, *Lachnospira*, *Roseburia*, and *Faecalibacterium* that correlated with increased fecal water. *Lachnospiraceae* belong to the core of gut microbiota among Firmicutes, colonizing the intestinal lumen from birth and increasing, in terms of species richness and their relative abundances during the host’s life [24]. Those species hydrolyze starch and other sugars to produce SCFAs, in particular, butyrate. A deeper understanding of the mechanisms involved in interactions with the host will represent the main future challenge, with a specific focus on the diet interactions stimulating or restricting their abundance and/or the presence of microbial pathways involved in the production of butyrate among other SCFA. The fact that improvement of transit characteristics and of constipation perception in middle-aged adults was better in the group treated with the higher dose of rhubarb extract, whereas changes in microbiota composition and fecal SCFA were more prominent in the group treated with the lower dose of rhubarb extract, may be attributed to other mechanisms previously described. Several studies have suggested that sennoside A, a major constituent of rhubarb, was responsible of its laxative effect [25,26,27]. Orally administered sennoside A is rarely absorbed in the small intestine; most of it reaches the colon to be metabolized into rheinanthrone by β-glucosidase of gut bacteria. Rheinanthrone might accelerate peristaltic movement in the colon, generating the laxative effect [26]. Besides the accelerated peristaltic movement, another pathway could be involved through aquaporin-3 -dependent inhibition of water absorption. Indeed, rheinanthrone may decrease aquaporin-3 expression in the colon to inhibit water transport from the luminal to the vascular side, leading to a laxative effect. This last mechanism may also explain why some subjects from Re × 2 group reported more diarrhea. Dose-dependency studies of rhubarb extract and/or its constituents on aquaporin-3 regulation could constitute a perspective of this work.

Further studies are needed to understand the potential impact of microbial-targeted therapies, including the modulation of *Lachnospiraceae,* with the end goal of their utilization in the prevention and treatment of intestinal diseases such as chronic constipation. 

We have identified several limitations to our study. First, increasing the number of participants would increase the statistical power of the analyses. Secondly, knowing that gender may impact outcomes related to constipation or the prevalence and severity of constipation, this parameter should be tested as possible effect modifiers. For example, enrolling a higher number of subjects but also controlling the power and the randomization for gender in a future study would be of interest [28]. In fact, research is needed to decipher whether gender and age differences may affect the symptoms of constipation, and how these covariates impact the prevalence and severity of constipation. Thirdly, gastrointestinal symptoms were not evaluated by using visual analogue scales (VAS) or by the Patient Assessment of Constipation Symptoms (PAC-SYM) questionnaire. Nevertheless, we can conclude that an increase in SCFA, notably by increasing butyrate-producing bacteria after a rhubarb extract supplementation, could be a promising strategy to relieve constipation with global impression of satisfaction about the management of transit in middle-aged people that experienced episodes of constipation.

## 4. Materials and Methods

### 4.1. Power Analysis and Sample Size

The sample size assumes a difference of 1 in the average change in stool frequency per week between the rhubarb groups and the placebo group. For the randomized, placebo-controlled study design and using the outcome measures described in the study protocol, the calculation performed (with Nquery Advisor software version 7.0, GraphPad Software DBA Statistical Solutions, San Diego, CA, USA (accessed on 24 May 2016)), with a standard deviation of 1, an increase of 0.5 stools per week in the placebo group and 1.5 stools in the rhubarb groups at risk alpha = 0.05, with a power of 80 and in a unilateral situation, gives a number of subjects to introduce of 13 per arm (45 subjects in total to take into account the loss of sight and the non-exploitable files).

### 4.2. Clinical Study

This prospective, randomized, double-blind, placebo-controlled study was designed according to the GCP (Good Clinical Practice) ICH E6 and was conducted at the CEN Experimental (Dijon, France), a private Clinical Investigation Center. The study was sponsored by Laboratoires Ortis (Elseborn, Belgium) and was monitored from March 2017 to March 2018 by the Contract Research Organization CEN (CEN Nutriment Business unit, Dijon, France) and was approved by the Comité de Protection des Personnes (Dijon, France; Approval Number: 2016-A00932-49—CEN 1486) prior to implementation. The trial study was listed on the NIH ClinicalTrials.gov website (NCT05541991). No changes were made to this trial after recruitment of the participants commenced. The author ensure that the study has been carried out in accordance with The Code of Ethics of the World Medical Association and followed the ethical guidelines set out in the Declaration of Helsinki. All participants provided written informed consent in compliance with the European law 2001/20/CE guidelines.

The allocation sequence to either placebo or study product was based on random sequence generated using MS EXCEL (simple randomization). The products were distributed to subjects in accordance with the randomization list. The randomization key, indicating to which products the batches given to patients corresponded, was kept in a sealed envelope. The coordinator and nurse enrolled participants and assigned them to placebo or rhubarb (computer-generated randomized numbers). Participants, care providers, researchers involved in data analyses were blinded to which arm participants were assigned. All information collected was kept in a secured area and sent for statistical analyses with a study number and without participant identifiers.

An overview of the study design is shown in Appendix A. Fifty-nine subjects were recruited. Among them, 17 subjects were excluded from the analysis because they did not meet the inclusion criteria. Forty-two received allocated intervention and completed the study (Appendix A). The inclusion criteria and exclusion criteria were given in Appendix A. Of note, knowing that physical exercise may affect gut transit [29], the volunteers were instructed not to change their physical activity habits during the study. The compliance with consuming the study product was 93.5%, 95.4% and 92.6% for placebo, Re × 1 and Re × 2, respectively.

The primary endpoint of this trial is to evaluate the effect of rhubarb extract supplementation on intestinal transit of subjects. The secondary endpoints are to evaluate changes in stool appearance, quality of life, relief and satisfaction of participants, changes in food consumption (survey with items according to recommendations of French National Nutrition and Health Program) and changes in gut composition.

### 4.3. Intervention

The subjects received a box with 90 tablets of either rhubarb root extract (Gutisrheum^®^, Laboratoires Ortis, Elsenborn, Belgium) or placebo in order to take daily 2 tablets every night at bedtime with a glass of water. Placebo tablets contained only excipients. Tablets were prepared by Laboratoires Ortis (Belgium) and were strictly identical in appearance. Subjects received 90 tablets in total but only 60 were required to complete the study; unused bags were returned to measure compliance (calculation: (90 − number of bags that were returned) × 100/60).

### 4.4. Transit Characteristics

The stool frequency and consistency were investigated through daily self-assessment using the Bristol Stool Form Scale (BSFS) [30], a simple tool for estimating intestinal transit time, from 2 weeks before the enrolment visit (run-in period) until the end of the intervention. The BSFS classifies stools into seven categories, including type 1, separate hard lumps, like nuts; type 2, sausage-shaped, but lumpy; type 3, like a sausage but with cracks on the surface; type 4, like a sausage or snake, smooth and soft; type 5, soft blobs with clear-cut edges; type 6, fluffy pieces with ragged edges, a mushy stool; type 7, watery, no solid pieces [31]. These types are categorized into slow transit (types 1 and 2), normal transit (types 3–5), and fast transit (types 6 and 7).

A questionnaire was submitted at baseline (d-14) and after intervention (d30) in order to evaluate the number of days with abdominal pain and intestinal discomfort evolution (accompanied or not with symptoms of diarrhea, bloating or gas). Patient Global Impression of Improvement (PGI-I) survey was also submitted at the end of the treatment (day 30); the PGI-I is a 1-item questionnaire asking to rate the perceived change about transit speed, transit problems and difficulty in evacuating in response to intervention at endpoint [32].

### 4.5. Plasma Analysis

The blood samples collected before (d-14) and after intervention (d30) were immediately centrifuged and plasma was transferred and kept at −20 °C until analysis. Blood and biochemical parameters were measured using standard laboratory techniques. Plasma cytokines (interleukin (IL)-1β, IL-8, IL-10, monocyte chemotactic protein-1 (MCP-1), interferon (IFN) γ and tumor necrosis factor (TNF) α) were determined in duplicate by multiplex immunoassays (Millipore, Belgium) and measured using LUMINEX xMAP technology (Biorad, Nazareth, Belgium) following the manufacturer’s instructions.

### 4.6. Quality of Life Assessment

Mental and physical wellbeing were assessed before (d-14) and after intervention (d30) through the Short-Form 12-item (SF-12) questionnaire consisting of 12 questions relating to: physical health problems, bodily pain, general health perceptions, vitality (energy/fatigue), social functioning, role limitations and general mental health (psychological distress and psychological well-being) [33,34]. This instrument yields two summary scores: a Mental Component Score (MCS), and a Physical Component Score (PCS). The SF12 takes 2 min to administer and has been validated for use with elderly people.

### 4.7. Gut Microbiota Analyses

Stool samples were collected at baseline and at the end of the 30 days of intervention (one fecal sample collected and stored immediately at −20 °C before day 0 and before day 30 and frozen samples were transferred within 7 days to −80 °C) for the analysis of the gut microbiota composition. Genomic DNA was extracted from feces using a PSP spin stool plus DNA kit (Stratec Biomolecular, Berlin, Germany), according to the manufacturer’s instructions. 16S rRNA gene profiling, targeting V1–V3 hypervariable region and sequenced on Illumina MiSeq were performed as described previously [8].

### 4.8. Fecal Short Chain Fatty Acid

Concentration of SCFA was determined in fecal matter using a method earlier described [35].

### 4.9. Statistical Analysis

Data are expressed as medians ± interquartile ranges for figures and as means ± SD or means ± SEM elsewhere. Between group differences were analyzed by Fisher or McNemar tests for categorical variables or Kruskal–Wallis test (followed by Dunn’s multiple comparisons test) for continuous variables. Within group analyses were evaluated using a Wilcoxon paired test (from baseline to 30 days of intervention). Mixed model ANOVA by Tukey’s multiple comparisons test were performed to compare effects over time. A significance level of *p* < 0.05 was used for all the analyses. For gut microbiota analysis, *p*-values of within group comparisons were corrected to control for the false discovery rate (FDR) for multiple tests according to the Benjamini and Hochberg procedure and a significance level of *q* < 0.05 was used. All analyses were conducted with, Graphpad Prism software version 9 (San Diego, CA, USA; www.graphpad.com (accessed on 26 January 2022)) except for the gut microbiota analysis where we used R version 3.5.2 (CRAN hosted by the Institute for Statistics and Mathematics of Wirtschaftsuniversität Wien, Austria).

## Figures and Tables

**Figure 1 ijms-23-14685-f001:**
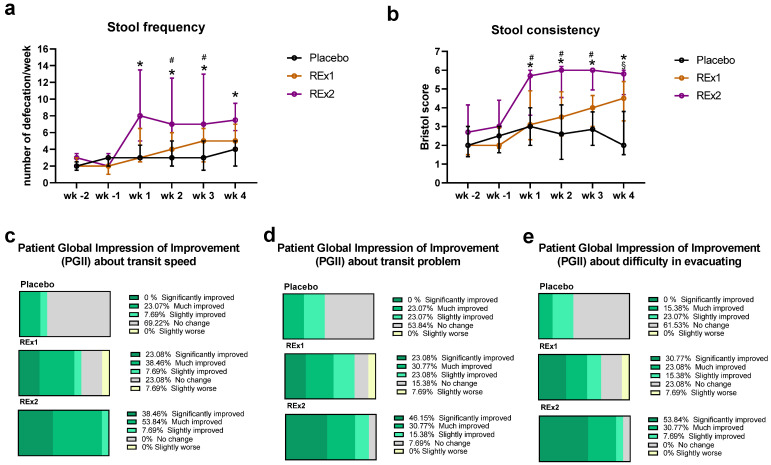
Transit characteristics in middle-aged adults receiving placebo or rhubarb extract (Re × 1 and Re × 2) for 30 days. Stool frequency (**a**), stool consistency (**b**) and perceived change of transit speed (**c**), transit problem (**d**) and difficulty in evacuating (**e**). Data are medians ± interquartile ranges for (**a**,**b**) (*n* = 13). Mixed models with multiple comparisons were performed to compare evolution of stool frequency (**a**) and consistency (**b**). * *p* < 0.05 Re × 2 vs. placebo; # *p* > 0.05 Re × 2 vs. Re × 1; § *p* < 0.05 Re × 1 vs. placebo. wk, week (wk -2: d-13 to d -7; wk -1: d-6 to d0; wk 1: d1 to d7; wk 2: d8 to d14; wk 3: d15 to d21; wk 4: d22 to d28).

**Figure 2 ijms-23-14685-f002:**
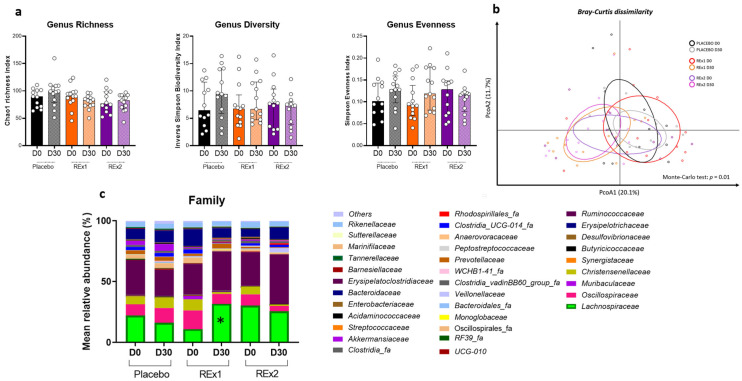
Gut microbiota composition of fecal matter from middle-aged adults at baseline or after receiving placebo or rhubarb extract (Re × 1 and Re × 2) for 30 days (*n* = 12 or 13). Alpha-diversity indexes related to bacterial richness or evenness (**a**). Principal coordinates analysis (PCoA) of the β-diversity indexes Bray–Curtis (**b**). Bar plots of relative abundance of family levels accounting for more than 1% (**c**). Data are medians ± interquartile ranges for (**a**). Matched-pairs Wilcoxon signed-rank tests were performed to compare changes from baseline (within-group variations; * *q* < 0.05). For β-diversity index, a Monte Carlo rank test was performed.

**Figure 3 ijms-23-14685-f003:**
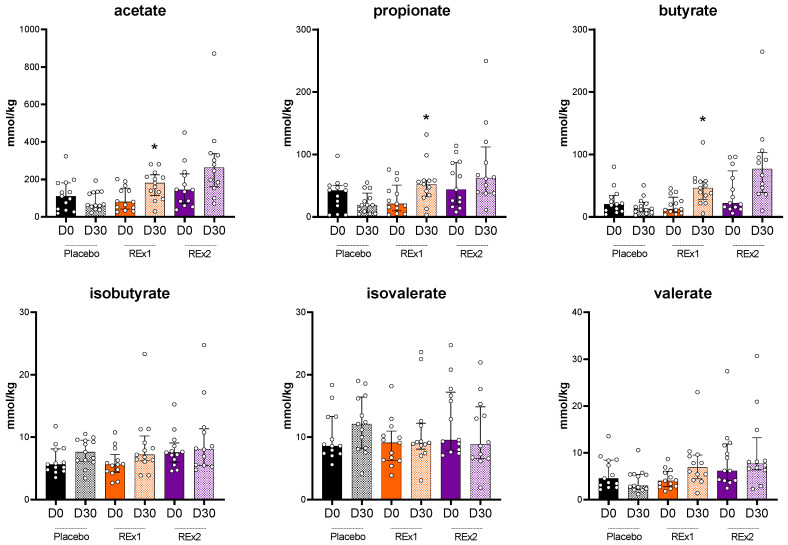
Short chain fatty acid composition of fecal dry matter from middle-aged adults after receiving placebo or rhubarb extract (Re × 1 and Re × 2) for 30 days (*n* = 13). Data are medians ± interquartile ranges. Baseline data were analyzed by Kruskal–Wallis test (*p* > 0.05). Within-group variations analyzed by matched-pairs Wilcoxon signed-rank tests to compare changes from baseline (* *p* < 0.05).

**Figure 4 ijms-23-14685-f004:**
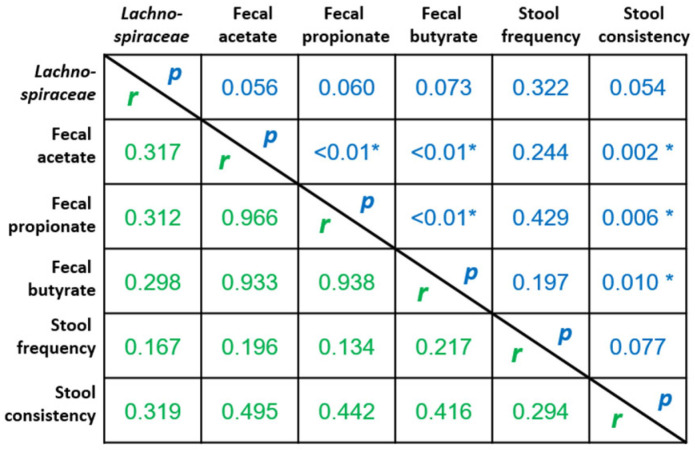
Correlation analysis between the changes in parameters that were significantly increased and/or improved due to the rhubarb extract supplementation (delta values from day 30 vs. baseline taking into account all subjects). Values are Spearman r coefficients (in green) and statistical *p* values (in blue) (*n* = 37, * *p* < 0.05).

**Table 1 ijms-23-14685-t001:** Baseline characteristics of participants ^1^.

	Placebo (*n* = 13)	Re × 1 (*n* = 13)	Re × 2 (*n* = 13)
Women/men N	13/0	12/1	12/1
Age (years old)	57 ± 6	58 ± 6	58 ± 6
Body weight (kg)	65.1 ± 9.1	61.2 ± 10.7	68.1 ± 10.1
BMI (kg/m2)	24.0 ± 3.7	22.6 ± 2.9	25.4 ± 3.6
SBP (mm Hg)	117 ± 13	113 ± 10	114 ± 16
DBP (mm Hg)	75 ± 10	71 ± 7	75 ± 10
ALAT (UI/L)	21.6 ± 7.6	18.2 ± 7.3	20.7 ± 13.1
ASAT (UI/L)	22.7 ± 4.8	19.9 ± 4.4	21.9 ± 3.8
Glycemia (g/l)	0.94 ± 0.16	0.92 ± 0.08	0.92 ± 0.06
Triglycerides (g/l)	0.92 ± 0.54	0.78 ± 0.23	0.92 ± 0.23
Cholesterol (g/l)	2.28 ± 0.34	2.04 ± 0.34	2.25 ± 0.25

^1^ Values are means ±SD. Baseline data were analyzed by Kruskal–Wallis test for continuous variables (*p* > 0.05) and Fisher test for categorical variables (ratio women/men; *p* > 0.05). ALAT, alanine aminotransferase; ASAT, aspartate aminotransferase; BMI, body mass index; DBP, diastolic blood pressure; SBP, systolic blood pressure.

**Table 2 ijms-23-14685-t002:** Safety parameters measured at baseline and after 30 days of treatment ^1^.

	Placebo (*n* = 13)		Re × 1 (*n* = 13)		Re × 2 (*n* = 13)	
	D0	D30	D0	D30	D0	D30
	65.1 ± 9.1	65.1 ± 9.0	61.2 ± 10.7	61.3 ± 10.5	68.1 ± 10.1	68.2 ± 10.2
Systolic blood pressure (mm Hg)	117.3 ± 12.5	115.4 ± 13.0	112.5 ± 9.6	109.2 ± 14.0	114.2 ± 15.9	114.8 ± 15.7
Diastolic blood pressure (mm Hg)	75.0 ± 9.6	74.6 ± 7.5	70.9 ± 6.7	68.5 ± 6.6	75.0 ± 9.6	71.5 ± 11.4
Alanine aminotransferase activity (U/L)	21.6 ± 7.6	20.2 ± 5.1	18.2 ± 7.3	18.0 ± 8.4	20.7 ± 13.1	19.9 ± 7.8
Aspartate aminotransferase activity (U/L)	22.7 ± 4.8	22.3 ± 5.1	19.9 ± 4.4	21.9 ± 5.6	21.9 ± 3.8	21.6 ± 3.4
Triglycerides (g/L)	0.92 ± 0.54	0.87 ± 0.45	0.78 ± 0.23	0.85 ± 0.48	0.92 ± 0.23	0.96 ± 0.30
Total cholesterol (g/L)	2.28 ± 0.34	2.18 ± 0.25	2.04 ± 0.34	2.06 ± 0.36	2.25 ± 0.25	2.28 ± 0.34
HDL-cholesterol (g/L)	0.65 ± 0.18	0.63 ± 0.17	0.61 ± 0.16	0.61 ± 0.16	0.56 ± 0.08	0.56 ± 0.08
LDL-cholesterol (g/L)	1.44 ± 0.30	1.38 ± 0.24	1.28 ± 0.32	1.27 ± 0.33	1.50 ± 0.21	1.53 ± 0.26
Glucose (g/L)	0.94 ± 0.16	0.93 ± 0.13	0.92 ± 0.08	0.87 ± 0.12	0.92 ± 0.06	0.92 ± 0.09
Potassium (mmol/L)	4.13 ± 0.24	4.19 ± 0.28	4.27 ± 0.46	4.08 ± 0.39	4.14 ± 0.24	4.11 ± 0.34
Hemoglobin (g/dL)	13.6 ± 0.9	13.5 ± 0.9	13.5 ± 1.1	13.1 ± 1.1	13.9 ± 1.2	13.6 ± 1.0
Leucocytes (G/L)	5.91 ± 2.00	5.26 ± 1.75 *	5.80 ± 1.15	5.92 ± 0.86	5.45 ± 1.27	5.30 ± 1.03
Lymphocytes (G/L)	1.83 ± 0.49	1.76 ± 0.39	1.82 ± 0.45	1.79 ± 0.39	1.94 ± 0.53	2.00 ± 0.48
Monocytes (G/L)	0.49 ± 0.13	0.45 ± 0.13	0.46 ± 0.08	0.49 ± 0.13	0.47 ± 0.14	0.46 ± 0.10
TNFα (pg/mL)	31.0 ± 33.8	35.9 ± 24.7	33.8 ± 22.3	41.7 ± 25.9	44.8 ± 34.2	55.4 ± 43.0
MCP1 (pg/mL)	197.6 ± 55.5	207.4 ± 62.5	215.0 ± 67.9	249.9 ± 103.6 *	277.9 ± 92.1	273.2 ± 63.8
IL1β (pg/mL)	11.3 ± 17.9	15.8 ± 12.8	12.6 ± 12.0	16.2 ± 12.4	16.4 ± 17.7	22.1 ± 22.5
IL8 (pg/mL)	5.9 ± 4.7	7.1 ± 3.9	6.7 ± 4.1	8.2 ± 5.2	8.7 ± 6.6	10.4 ± 6.9
IL10 (pg/mL)	29.3 ± 53.8	29.2 ± 35.6	23.3 ± 26.0	30.2 ± 25.7	36.2 ± 40.7	49.3 ± 57.1
IFNγ (pg/mL)	27.4 ± 40.3	38.7 ± 30.4	29.9 ± 27.4	43.6 ± 37.5	48.2 ± 58.2	60.9 ± 56.2

^1^ Values are means ±SD. IL, Interleukin; IFNγ, Interferon-gamma; MCP-1 Monocyte chemotactic protein-1; TNFα, tumor necrosis factor-alpha. Data are means ± SD. Matched-pairs Wilcoxon signed-rank tests were performed to compare changes from baseline (Within-group variations; * *p* < 0.05).

## Data Availability

Data described in the manuscript will be made available upon reasonable request from the corresponding author.

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
