# Peer review of "Constipation Mitigation by Rhubarb Extract in Middle-Aged Adults Is Linked to Gut Microbiome Modulation: A Double-Blind Randomized Placebo-Controlled Trial"

_ijms, 2022, doi:10.3390/ijms232314685_

Round 1
Reviewer 1 Report
Neyrinck et al. investigated the influence of rhubarb extract (RE) supplementation on constipation in middle-aged volunteers (primarily women) through a double-blind, randomized placebo-controlled trial. Two different doses of RE were chosen (12.5 and 25 mg per day based on rhein content for the REx1 and REx2 groups, respectively) with maltodextrin as the placebo over the course of 30 days. Patients perceived a dose-responsive improvement in their transit speed, transit problem, and evacuation difficulty. Intervention may have improved patient mental wellbeing. Stool frequency and consistency improved for the REx2 group after intervention, although there was a significant increase in complaints of diarrhea or loose stools. The REx1 group saw improved stool consistency after 4 weeks of treatment. 16S sequencing was performed, with the authors claiming an increase in Lachnospiraceae observed as a result in treatment in the REx1 group. Short-chain fatty acid analyses were also performed, with significant increases in acetate, propionate, and butyrate observed in the REx1 group.
The mechanism of RE alleviating constipation through increasing Lachnospiraceae as presented is questionable. In Figure 2c it appears that a significant increase in Lachnospiraceae is an artifact of the REx1 group starting out with the lowest D0 level and ending up with the highest D30. Highlighting this is that the REx2 D0 level is nearly as high as the REx1 D30. Is the REx1 D30 increase still significant when compared to the average of all R0 groups? Furthermore, can the authors identify and present more specific changes in the microbiota that may account for the increase in SCFAs? There are numerous examples of RE / anthraquinones modulating microbial composition and function in the literature, how does this study compare?
Figure 3 would be better presented as bar graphs that include the baseline values as well with each data point shown to get an idea of how the data is distributed (similar to presentation in Figure 2a). Are the values presented relative to the D0 in each group, or an averaged D0 across all groups? As with the microbiota composition data, if there are notable differences in the D0 across groups that bring significance into question, it would be appropriate to include comparisons against a D0 containing values from all groups.
Why does the higher RE treatment show the greater improvements in constipation (Figure 1), but does not show a significant change in microbiota composition and SCFA production? What other mechanisms may the RE treatment work through?
Figure 1b is somewhat cluttered with each R0 group shown separately, consider grouping all the R0 data to improve clarity.
Please provide more detail regarding Figure 4, is this data representing all groups?
Why does the REx2 group report more diarrhea / loose stools?
Author Response
Dear Reviewer
We thank you for your useful comments. We have performed several changes in the text in order to answer the comments and to meet your suggestions. The paper with track changes indicates where those changes have been implemented.

Reviewer 2 Report
The article offers new insights into the influence of rhubarb supplementation on chronic constipation relief in middle-aged adults connected to gut microbiota composition. The authors present interesting observations, and the article is well-written. The introduction was comprehensive and informative. Results are clearly described and illustrated. The discussion is satisfactory.
I recommend the article for publication.
Author Response
We thank the reviewer for its comment

Reviewer 3 Report
The manuscript entitled "Constipation mitigation by rhubarb extract in middle-aged adults is linked to gut microbiome modulation: a double-blind randomised placebo-controlled trial" is a report of a clinical investigation using a Rhubarb extract to treat chronic constipation. The novelty of this study is that it utilises rhubarb in a human trial as opposed to the rodent model. Consequently it provides data that is necessary for justifying its use as an alternative to commonly used medications, but also as an adjunctive therapy for the reduction of medication use.
The manuscript is very well prepared and does not require English modifications nor modifications. It is therefore recommended for publication.
Author Response
We thank the reviewer for its comment

Round 2
Reviewer 1 Report
Thank you for the informative and thoughtful responses. This is great work and a very nice manuscript.